

# Investigating the impact of vulnerability datasets on deep learning-based vulnerability detectors

Lili Liu*, Zhen Li*, Yu Wen and Penglong Chen

School of Cyber Security and Computer, Hebei University, Baoding, Hebei Province, China
* These authors contributed equally to this work.

## ABSTRACT

Software vulnerabilities have led to system attacks and data leakage incidents, and software vulnerabilities have gradually attracted attention. Vulnerability detection had become an important research direction. In recent years, Deep Learning (DL)-based methods had been applied to vulnerability detection. The DL-based method does not need to define features manually and achieves low false negatives and false positives. DL-based vulnerability detectors rely on vulnerability datasets. Recent studies found that DL-based vulnerability detectors have different effects on different vulnerability datasets. They also found that the authenticity, imbalance, and repetition rate of vulnerability datasets affect the effectiveness of DL-based vulnerability detectors. However, the existing research only did simple statistics, did not characterize vulnerability datasets, and did not systematically study the impact of vulnerability datasets on DL-based vulnerability detectors. In order to solve the above problems, we propose methods to characterize sample similarity and code features. We use sample granularity, sample similarity, and code features to characterize vulnerability datasets. Then, we analyze the correlation between the characteristics of vulnerability datasets and the results of DL-based vulnerability detectors. Finally, we systematically study the impact of vulnerability datasets on DL-based vulnerability detectors from sample granularity, sample similarity, and code features. We have the following insights for the impact of vulnerability datasets on DL-based vulnerability detectors: (1) Fine-grained samples are conducive to detecting vulnerabilities. (2) Vulnerability datasets with lower inter-class similarity, higher intra-class similarity, and simple structure help detect vulnerabilities in the original test set. (3) Vulnerability datasets with higher inter-class similarity, lower intra-class similarity, and complex structure can better detect vulnerabilities in other datasets.

## INTRODUCTION

Software vulnerabilities refer to specific flaws in software that enable attackers to carry out malicious activities. The threat of system attacks and data leakage make software security vulnerabilities a vital issue (*Chen et al., 2019*; *Zhu et al., 2017*; *Zhu et al., 2020*). Using source code is effective in open-source code vulnerability detection because it can uncover vulnerabilities from a root cause. Source code static vulnerability detection involves code similarity-based methods (*Kim et al., 2017*; *Li et al., 2016*; *Jang, Agrawal & Brumley, 2012*)

Corresponding authors
Lili Liu, liulili2847@gmail.com
Zhen Li, lizhenhbu@gmail.com

and pattern-based methods (*Yamaguchi et al., 2013*; *Neuhaus et al., 2007*; *Yamaguchi, Lottmann & Rieck, 2012*; *Grieco et al., 2016*). Code similarity-based methods have a high rate of false positives and false negatives (*Johnson et al., 2013*). Pattern-based methods include rule-based methods and machine learning-based methods. Machine learning-based methods include traditional machine learning-based methods and Deep Learning (DL)-based methods. Rule-based methods and traditional machine learning-based methods rely on experts to manually extract features (*Zhen et al., 2018*). The DL-based method does not require a manual definition of features and has a low rate of false negatives and false positives.

In this article, we studied DL-based vulnerability detectors. DL methods automatically capture and determine features from the training set and then learn to identify vulnerabilities. DL-based vulnerability detectors rely on vulnerability datasets. This paper explored C/C++ vulnerability datasets. The existing C/C++ vulnerability datasets mainly include artificially synthesized data (*Black, 2018*; *Zhen et al., 2018*; *Li et al., 2021c*), artificially modified data (*Booth, Rike & Witte, 2013*; *Zhen et al., 2018*; *Li et al., 2021c*), and real-world open-source code (*Russell et al., 2018*; *Fan et al., 2020*; *Wang et al., 2020*; *Zhou et al., 2019*; *Lin et al., 2019a*).

A recent study (*Chakraborty et al., 2020*) found that the authenticity, imbalance, and repetition rate of the vulnerability dataset will affect the results of the DL-based vulnerability detector. However, the existing research on vulnerability datasets is not comprehensive. There is no characterization of vulnerability datasets or systematic evaluation of the impact of vulnerability datasets on DL-based vulnerability detectors.

## Challenges

The major challenges of investigating the impact of vulnerability datasets on DL-based vulnerability detectors are as follows:

- The challenge of characterizing vulnerability datasets. Vulnerability datasets are different from text, image, and other datasets. The internal structure of the code in the dataset is more complex. It is characterized by a very abstract concept that is difficult to represent with intuitive data.
- The challenge of vulnerability dataset evaluation methods. The criterion for evaluating the quality of a vulnerability dataset is its impact on the results of the vulnerability detector. However, for the same vulnerability dataset, using different DL-based vulnerability detectors generates significantly different results. Therefore, it is also difficult to study the quality of the dataset by stripping the impact of the performance of DL-based vulnerability detectors.

## Contributions

We characterized vulnerability datasets according to three aspects: sample granularity, sample similarity, and code features. We studied the impact of C/C++ vulnerability datasets on DL-based vulnerability detectors to obtain insights. Based on these insights, we

provide suggestions for the creation and selection of vulnerability datasets. Our main contributions are as follows:

- We proposed methods to characterize the sample similarity and code features. We calculated the distance between the sample vectors to obtain the inter-class and intra-class distance. We used the inter-class and intra-class distance to express the similarity between the classes and the similarity across the class of samples. We selected five features to characterize code and measure sample complexity, sample size, and subroutine call-related information.

- We used sample granularity, sample similarity, and code features to characterize vulnerability datasets. Then we analyzed the characteristics of vulnerability datasets and the results of DL-based vulnerability detectors to study the impact of the vulnerability datasets on DL-based vulnerability detectors.

- We selected four vulnerability datasets, three methods of representation, and four DL-based vulnerability detectors for experiments. We found that the sample granularity, sample similarity, and code features of the dataset impacted DL-based vulnerability detectors in the following ways: (1) Fine-grained samples were conducive to detecting vulnerabilities; (2) vulnerability datasets with higher inter-class similarity, lower intra-class similarity, and simple structure were conducive to detecting vulnerabilities in the original test set; and (3) vulnerability datasets with lower inter-class similarity, higher intra-class similarity, and complex structure helped detect vulnerabilities in other datasets.

## RELATED WORK

This article studies the impact of the vulnerability dataset on DL-based vulnerability detectors. The following is related information on three aspects: vulnerability detectors, vulnerability datasets, and research on vulnerability datasets.

### Vulnerability detectors

Source code vulnerability detection involves methods based on code similarity (*Kim et al., 2017*; *Li et al., 2016*; *Jang, Agrawal & Brumley, 2012*) as well as pattern-based methods (*Yamaguchi et al., 2013*; *Neuhaus et al., 2007*; *Yamaguchi, Lottmann & Rieck, 2012*; *Grieco et al., 2016*). Code similarity-based methods can detect vulnerabilities due to code cloning, but these methods have a high false-positive rate and false-negative rate (*Johnson et al., 2013*). The rule-based method is a pattern-based method that relies on experts manually extracting features. Machine learning-based methods are also pattern-based methods that include traditional machine learning-based methods and DL-based methods. Traditional machine learning-based methods also rely on experts to manually extract features. The method of manually extracting features is time-consuming and laborious, and it is not easy to entirely extract the features (*Zhen et al., 2018*). DL-based methods automatically extract features with a low rate of false positives and false positives. DL-based vulnerability detection methods are divided into the following three types according to their feature extraction method.

The first type is sequence-based vulnerability detection. This type of research uses Deep Neural Networks (DNNs) to extract feature representations from sequence code entities, mainly text sequence and function call sequence. The text sequence mainly contains source code text (*Sestili, Snavely & VanHoudnos, 2018*; *Choi et al., 2017*; *Peng et al., 2015*; *Li et al., 2021b*), assembly instructions (*Le et al., 2019*), and source code processed by the code lexer (*Russell et al., 2018*). The function call sequence includes static calls and dynamic calls (*Grieco et al., 2016*; *Wu et al., 2017*). Additionally, they allow neural networks to capture flow-based patterns and advanced features (*Zhen et al., 2018*; *Zou et al., 2019*; *Li et al., 2021c*; *Cheng et al., 2021*).

The second type is Abstract Syntax Tree (AST)-based code vulnerability detection. AST retains the hierarchical structure of sentence and expression organization. It contains a relatively large amount of code semantics and syntax. Therefore, AST can be a valuable source for learning feature representations related to potentially vulnerable patterns. This type of method first extracts the ASTs of the code and then combines them with the seq2seq (*Dam et al., 2017*), bidirectional long short-term memory (BLSTM) (*Farid et al., 2021*; *Lin et al., 2018*), or other networks (*Wang, Liu & Tan, 2016*; *Lin et al., 2017*) to extract the semantic features of the code.

The third type is graph-based vulnerability detection. These studies use DNN to learn feature representations from different types of graph-based program representations, including AST, Control Flow graphs (CFGs), Program Dependency graphs (PDGs), data-dependent graphs (DDGs), and combinations of these graphs (*Shar & Tan, 2013*; *Duan et al., 2019*; *Dong et al., 2018*; *Harer et al., 2018*; *Lin et al., 2019b*) as input to DNNs for learning deep feature representations. Based on this, some studies have used multiple types of composite graphs to express richer semantic information (*Zhou et al., 2019*; *Chakraborty et al., 2020*; *Wang et al., 2020*). This paper focuses on these three types of DL-based vulnerability detectors.

## Vulnerability datasets

The existing C/C++ vulnerability datasets are mainly divided into the following three types according to the collection method. The first type is artificially synthesized data using known as vulnerability patterns, such as Software Assurance Reference Dataset (SARD) (*Black, 2018*; *Zhen et al., 2018*; *Li et al., 2021c*). This type of data is relatively simple and has a single vulnerability pattern. The second type is original data that has been manually modified, such as National Vulnerability Dataset (NVD) (*Booth, Rike & Witte, 2013*; *Zhen et al., 2018*; *Li et al., 2021c*; *Bhandari, Naseer & Moonen, 2021*) and other vulnerability databases. They annotate and modify the collected data to indicate vulnerabilities. The third type are real-world open-source datasets, such as open-source repositories like GitHub (*Russell et al., 2018*; *Fan et al., 2020*; *Wang et al., 2020*; *Wang et al., 2021*) and open-source software (*Zhou et al., 2019*; *Lin et al., 2019a*; *Zheng et al., 2021*). This type of data involves a wide range of vulnerabilities and different structures, reflecting real-world software vulnerabilities. Generally, the unpatched version is regarded as vulnerability data, and the patched version is regarded as non-vulnerability data. This paper studies these three types of C/C++ vulnerability datasets.

### Research on vulnerability datasets

Previous studies have explored which code changes are more prone to contain vulnerabilities in datasets (*Bosu et al., 2014*), and have analyzed the dependencies between vulnerabilities (*Li et al., 2021a*) and the vulnerability distribution (*Liu et al., 2020*). A recent study (*Chakraborty et al., 2020*) calculated the authenticity of existing vulnerability datasets, the proportion of data with and without vulnerabilities, and repeated samples. That study also found that low authenticity is not conducive to detecting real-world vulnerabilities. Unbalanced datasets usually make DL-based vulnerability detectors ineffective. Vulnerability datasets with high repeated sample rates may help detect certain vulnerabilities but not others. The current research does not characterize vulnerability datasets or the impact of the vulnerability dataset on DL-based vulnerability detectors.

## DESIGN

The purpose of this paper is to study the impact of vulnerability datasets on DL-based vulnerability detectors. This paper explores the impact of vulnerability datasets on DL-based vulnerability detectors from three aspects: granularity, similarity, and code features. The following are our research motivations:

### Granularity

When we studied at SySeVR (*Li et al., 2021c*), we found that there were differences in the results obtained at the slice-level dataset and function-level dataset. And the slicing technology extracts the information related to the vulnerability. Therefore, we believe that granularity will have an impact on the results of vulnerability detectors and do research on granularity in this paper.

### Similarity

Vulnerability detection is a binary classification problem of deep learning. The deep learning model learns the characteristics of the two types of samples through vectors. So the inter- and intra-class similarity of input vectors will affect the learning of the deep learning model. If the two classes of vectors are very different, it is easier to learn discriminative features. If the same-class vector differences are small, it is easier to learn the features of each class. The vectors come from samples, and the difference between the vectors is not only the representation method but also the difference of samples. Therefore, we believe that the similarity between the samples themselves may have an impact on the effectiveness of vulnerability detectors. In this paper, we investigate the effect of inter-class similarity and intra-class similarity on vulnerability detector performance.

### Code features

A study (*Chakraborty et al., 2020*) found that the effects of artificially synthesized datasets and real-world datasets showed differences. We argue that the difference between synthetic and real datasets lies not in their origin, but code features, such as code complexity. Therefore, this paper comprehensively analyzes the code features and studies the impact of code features on the effect of vulnerability detectors.

We achieved insights by answering the following research questions:

*RQ1: How does the granularity of vulnerability dataset samples affect DL-based vulnerability detectors?* The sample granularity of vulnerability datasets is mainly divided into function level and slice level (*Li, Wang & Nguyen, 2021*). The source code files are divided by function at the function level and labelled as vulnerable or non-vulnerable. In the slice set, the slices are labelled according to whether there are vulnerable lines in the slices. We processed the same dataset into different granularities. Then, we studied the impact of sample granularity on DL-based vulnerability detectors.

*RQ2: How does the similarity of vulnerability dataset samples affect DL-based vulnerability detectors?* Vulnerability datasets are divided into two categories: vulnerable data and non-vulnerable data. We consider that sample similarity has two aspects: inter-class similarity and intra-class similarity. Inter-class similarity refers to the similarity between two classes of samples. Intra-class similarity refers to the similarity between samples of the same class. We studied the impact of sample similarity between classes and within classes on DL-based vulnerability detectors from the vectors.

*RQ3: How do the code features of vulnerability dataset samples affect DL-based vulnerability detectors?* The code sample has features that cannot be expressed intuitively, such as code complexity, code amount, and subroutine call-related information. We characterized code features by selecting five features for comprehensive analysis and measured sample complexity, sample size, and subroutine call-related information. Then, we studied the impact of code features on DL-based vulnerability detectors.

To study the impact of the vulnerability dataset on the DL-based vulnerability detectors from the above three aspects, we first extracted the characteristics of vulnerability datasets to characterize the vulnerability datasets. Then, we used vulnerability datasets to train DL-based vulnerability detectors and get the test results of DL-based vulnerability detectors. We then performed association analysis on the characteristics of the vulnerability datasets and the results of DL-based vulnerability detectors. Finally, we gain insights into the impact of vulnerability datasets on DL-based vulnerability detectors from sample granularity, sample similarity, and code features. Figure 1 briefly introduces the main research process of this paper. We used steps I–III to characterize the vulnerability dataset according to these three aspects: sample granularity, sample similarity, and code features.

## STEP I: Generating code samples

This step was to adapt the vulnerability dataset to the input format requirements of the vulnerability detector. We generated a set of vulnerable code samples $A = \{A_1, A_2,..., A_m\}$ and a set of non-vulnerable code samples $B = \{B_1, B_2,..., B_n\}$ from the vulnerability dataset, where $m$ and $n$ were the number of vulnerable code samples and non-vulnerable code samples. We generated code samples with two granularities: function-level and slice-level.

When generating function-level code samples, we divided the code files into function units. Then, we labelled the functions according to the information provided by the vulnerability dataset. The labelling method should be determined according to the

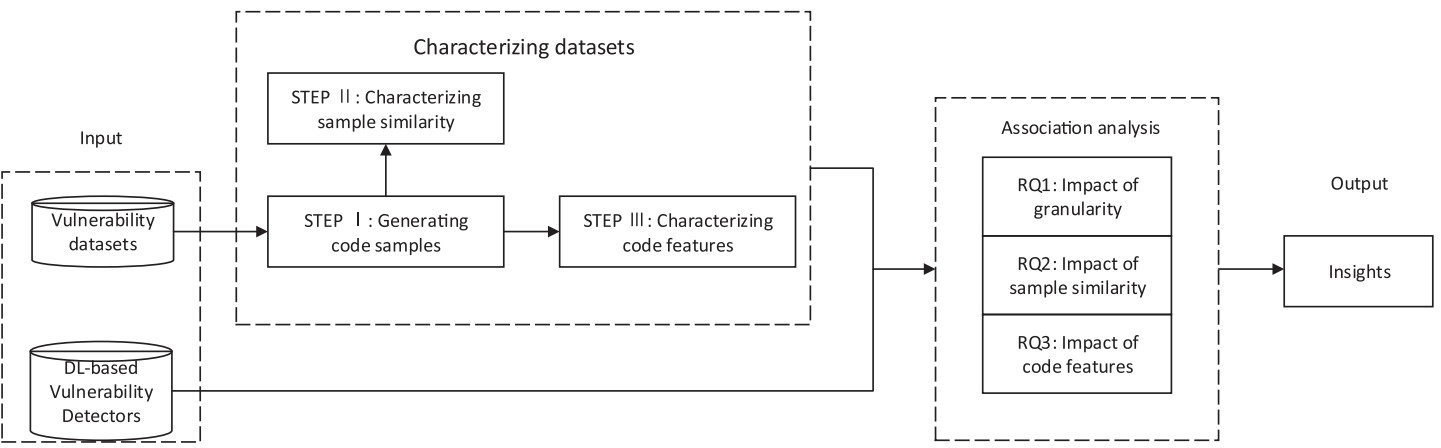

**Figure 1 The overview of this paper: vulnerability datasets and DL-based vulnerability detectors as input.** Steps I–III characterized the characteristics of vulnerability datasets. The characteristics of vulnerability datasets and the results of DL-based vulnerability detectors were used for association analysis to obtain the answers to RQ1-3. Insights were achieved through the above analysis.

requirements of the vulnerability detector. Usually, vulnerable functions are labelled "1", and non-vulnerable functions are labelled "0".

When generating slice-level code samples, the first step is determining whether the vulnerability dataset contains vulnerability line information. This is because we needed vulnerability line information to determine whether the slice contained vulnerability data, and then we could label slices. For a vulnerability detector, only the labelled training set is meaningful. We then generated a PDG diagram of the source code and generated corresponding program slices for the code elements in the PDG. Finally, we labelled the slices according to the vulnerability line information provided by the vulnerability dataset. The slices containing the vulnerability lines were labelled as vulnerable data, and those that did not contain the vulnerability lines were labelled as non-vulnerable data.

### STEP II: Characterizing sample similarity

This step was to characterize the sample similarity of the vulnerability dataset. We considered two types of sample similarities: inter-class similarity and intra-class similarity. In order to enable the DL model to learn features better, it was necessary to simplify the complex original data and express the original data as vectors. We represented vulnerable code samples set $A$ and non-vulnerable code samples set $B$ as multi-dimensional vectors sets $V_A = \{V_{A1}, V_{A2}, \ldots, V_{Am}\}$ and $V_B = \{V_{B1}, V_{B2}, \ldots, V_{Bn}\}$.

The methods for representing codes as vectors were mainly divided into sequence-based, AST-based, and graph-based. Sequence-based representation means that the code is treated as text sequences, regardless of the internal structure of the code. For example, word2vec (*Mikolov et al., 2013*) encodes tokens in the code. AST-based representation is a tree representation of the abstract syntax structure of the source code. First, it decomposes the code into a set of paths in the corresponding AST. Then, it uses the neural network to learn the representation of each path and how to integrate the representation of all

paths. Abstract Syntax Tree Neural Network (ASTNN) (*Zhang et al., 2019*) and code2vec (*Alon et al., 2019*) are two representations based on AST. Graph-based representation is based on multiple graphs that explicitly encode different control dependencies and data dependencies as edges of heterogeneous graphs. This kind of representation is more concerned with control flow and data flow information, such as Gated Graph Neural Networks (GGNN) (*Zhou et al., 2019*).

After representing the code samples as vectors sets $V_A$ and $V_B$, we reduced their dimensions to two-dimensional vectors sets $v_A = \{v_{A1}, v_{A2},\ldots, v_{Am}\}$ and $v_B = \{v_{B1}, v_{B2},\ldots, v_{Bn}\}$. We represented the similarity between classes by the distance between classes. We calculated this by averaging the average distance between two types of samples, denoted by $D_{inter}$,

$$D_{inter} = \frac{1}{m * n}\sum_{i=1}^{m}\sum_{j=1}^{n}D(v_{Ai}, v_{Bj}). \tag{1}$$

$D(v_1, v_2)$ represents the cosine distance between $v_1$ and $v_2$ (*Chakraborty et al., 2020*),

$$D(v_1, v_2) = 1 - |\frac{v_1 \cdot v_2}{||v_1|| * ||v_2||}|. \tag{2}$$

We represent the similarity within the class by the distance within the class. We calculated the sum of the average distance between each type of sample, denoted by $D_{intra}$,

$$D_{intra} = \frac{1}{m^2}\sum_{i=1}^{m}\sum_{j=1}^{m}D(v_{Ai}, v_{Aj}) + \frac{1}{n^2}\sum_{i=1}^{n}\sum_{j=1}^{n}D(v_{Bi}, v_{Bj}). \tag{3}$$

The larger the $D_{inter}$ and $D_{intra}$, the lower the corresponding similarity.

We also used relative entropy to measure sample similarity. $En(a,b)$ represents the relative entropy between two samples a and b. The inter-class relative entropy is calculated as the average relative entropy between two types of samples, denoted as $En_{inter}$,

$$En_{inter} = \frac{1}{m * n}\sum_{i=1}^{m}\sum_{j=1}^{n}En(A_i, B_j). \tag{4}$$

The intra-class relative entropy is calculated as the sum of the average relative entropy between samples of each class, denoted as $En_{intra}$,

$$En_{intra} = \frac{1}{m^2}\sum_{i=1}^{m}\sum_{j=1}^{m}En(A_i, A_j) + \frac{1}{n^2}\sum_{i=1}^{n}\sum_{j=1}^{n}En(B_i, B_j). \tag{5}$$

The larger the $En_{inter}$ and $En_{intra}$, the lower the corresponding similarity.

### STEP III: Characterizing code features
This step was to characterize the code features of the vulnerability dataset, such as code complexity, sample size, and subroutine call-related characteristics. In order to

**Table 1  Code features of vulnerability datasets.**

| Feature | Description |
| --- | --- |
| AvgCyclomatic | Average cyclomatic complexity for all nested functions or methods |
| AvgEssential | Average Essential complexity for all nested functions or methods |
| AvgLine | Average number of lines for all nested functions or methods |
| AvgCountInput | Number of calling subprograms plus global variables read |
| AvgCountOutput | Number of called subprograms plus global variables set |

**Table 2  Summary of vulnerability datasets.**

| Dataset | Source | Category | Vulnerable samples | Non-vulnerable samples |
| --- | --- | --- | --- | --- |
| SySeVR | SARD + NVD | Synthesized, manually modified | 2,091 | 13,502 |
| FUNDED | GitHub | Open-source repository | 5,200 | 5,200 |
| Design | Qemu + FFMPeg | Open-source software | 10,067 | 12,294 |
| REVEAL | Chromium + Debian | Open-source software | 1,664 | 16,505 |

characterize the code features of vulnerability datasets listed above, we chose five features from SciTools: https://scitools.org.uk. *AvgCyclomatic, AvgEssential, AvgLine, AvgCountInput* and *AvgCountOutput*. Table 1 describes these features. *AvgCyclomatic* and *AvgEssential* are used to indicate the complexity of the code. *AvgLine* represents the sample size. *AvgCountInput* and *AvgCountOutput* are used to indicate the subroutine calls of the code.

# EXPERIMENTAL SETUP

## Implementation

We used Pytorch 1.4.0 with Cuda version 10.1 and TensorFlow 1.15 (or 1.12) to implement models. We ran our experiments on double Nvidia Geforce 2080Ti GPU, Intel (R) Xeon(R) 2.60 GHz 16 CPU. The time to train a single vulnerability detection model was between 4 h and 17 h.

## Vulnerability datasets

We choose four datasets to conduct experiments: SySeVR (*Li et al., 2021c*), FUNDED (*Wang et al., 2020*), Design (*Zhou et al., 2019*), and REVEAL (*Chakraborty et al., 2020*). Table 2 contains an overview of these four vulnerability datasets. Here are the reasons for choosing them:

- In order to study the impact of granularity on vulnerability detectors, it is necessary to generate slice-level and function-level data for the same vulnerability dataset. Therefore, we selected two vulnerability datasets containing vulnerability line information: SySeVR (*Li et al., 2021c*) and FUNDED (*Wang et al., 2020*).
- In order to study the impact of sample similarity and code features on vulnerability detectors, the difference of vulnerability datasets should be as large as possible.

Therefore, we chose vulnerability datasets from different sources. SySeVR (*Li et al., 2021c*) comes from SARD (*Black, 2018*) and NVD (*Booth, Rike & Witte, 2013*), and FUNDED (*Wang et al., 2020*) comes from GitHub. We then chose Devign (*Zhou et al., 2019*) from Qemu and FFMPeg as another dataset. We used the above three vulnerability datasets from different sources to train and test vulnerability detectors.

- We chose REVEAL (*Chakraborty et al., 2020*) as the public test set to ensure fairness. Since it is a real-world vulnerability dataset. Additionally, it is from a source different to the other three three datasets.

## Representation methods and DL-based vulnerability detectors

We chose three types of representation methods: sequence-based representation method - word2vec (*Mikolov et al., 2013*), AST-based representation method-code2vec (*Alon et al., 2019*), and graph-based representation method-GGNN (*Zhou et al., 2019*).

We chose four vulnerability detectors to conduct experiments: SySeVR (*Li et al., 2021c*), VulDeePecker (*Zhen et al., 2018*), REVEAL (*Chakraborty et al., 2020*), and C2V-BGRU. Here are the reasons for choosing them:

- To study the impact of granularity on vulnerability detectors, a vulnerability detector that can accept both function level and slice level as input should be selected. Therefore, we chose SySeVR (*Li et al., 2021c*) to study the impact of granularity on vulnerability detectors.

- In order to study the impact of sample similarity and code features, we needed vulnerability detectors that use the three representation methods studied in this paper and different DL models to avoid the bias caused by specific DL models. Therefore, we chose VulDeePecker (*Zhen et al., 2018*) based on word2vec (*Mikolov et al., 2013*) and BLSTM, REVEAL (*Chakraborty et al., 2020*) based on GGNN (*Zhou et al., 2019*) and MLP, and a variant of SySeVR (*Li et al., 2021c*) based on code2vec (*Alon et al., 2019*) and BGRU (called C2V-BGRU).

## Evaluation metrics

Our approaches were based on four popular evaluation metrics used for classification tasks: Accuracy (ACC), Precision (P), Recall (R), and F1-score (F1). Let True Positive (TP) be the number of samples with vulnerabilities detected correctly, True Negative (TN) be the number of samples with non-vulnerabilities detected correctly, False Positive (FP) be the number of samples with false vulnerabilities detected, and False Negative (FN) be the number of samples with true vulnerabilities undetected. Accuracy (ACC) indicates the proportion of all correctly classified samples to total samples, $ACC = (TP + TN)/(TP + TN + FP + FN)$. Precision (P), also known as the Positive Predictive rate, indicates the correctness of predicted vulnerable samples, $P = TP/(TP + FP)$. Recall (R) indicates the effectiveness of vulnerability prediction, $R = TP/(TP + FN)$. F1-score (F1) is defined as the geometric mean of Precision and Recall, $F1 = 2 * (P * R)/(P + R)$.

**Table 3 The result of SySeVR on function-level and slice-level vulnerability samples.**

| Dataset | Granularity | Accuracy (%) | Precision (%) | Recall (%) | F1-score (%) |
|---------|-------------|--------------|---------------|------------|--------------|
| FUNDED  | Function    | 72.34        | 57.02         | 55.73      | 56.38        |
|         | Slice       | 75.48        | 63.23         | 59.55      | 65.47        |
| SySeVR  | Function    | 80.36        | 85.13         | 82.52      | 83.37        |
|         | Slice       | 89.57        | 96.54         | 84.02      | 89.89        |

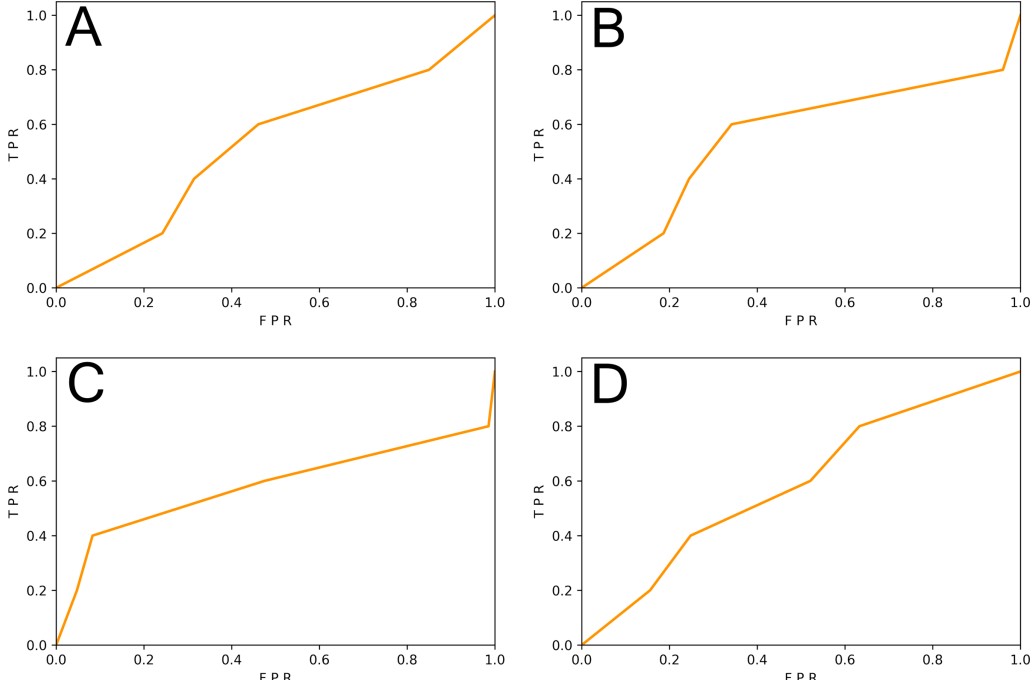

**Figure 2 (A–D) The ROC curve of three vulnerability detectors tested on the original test set.**

## EXPERIMENTAL RESULTS

### Impact of granularity (RQ1)

This subsection studied the impact of different sample granularity on the DL-based vulnerability detector. We generated slice-level samples and function-level samples of SySeVR (*Li et al., 2021c*) and FUNDED (*Wang et al., 2020*). Since SySeVR used the ratio of 80%:20% in original papers, we follow this ratio to ensure maximum restoration of vulnerability detectors. 80% of the samples were training set and 20% of the samples were test set. The training set was used to train the SySeVR (*Li et al., 2021c*) vulnerability detector. The function-level test set was used for testing. The results are shown in Table 3 and Fig. 2.

We observed that the F1-score of the vulnerability detector trained on the slice-level data of the SySeVR (*Li et al., 2021c*) dataset was 9.09% higher than that of the vulnerability detector trained on the function-level data. The F1-score of vulnerability detectors trained

**Table 4  The inter-class distance and intra-class distance of the three vulnerability datasets under the three representation methods.**

| Dataset | Representation | $D_{inter}$ | $D_{intra}$ | $En_{inter}$ | $En_{intra}$ |
|---|---|---|---|---|---|
| FUNDED | word2vec | 0.1507 | 0.3578 | 0.6739 | 0.5317 |
|  | code2vec | 0.3201 | 0.4854 |  |  |
|  | GGNN | 0.3205 | 0.5942 |  |  |
| SySeVR | word2vec | 0.4013 | 0.1327 | 0.8578 | 0.3790 |
|  | code2vec | 0.4945 | 0.2343 |  |  |
|  | GGNN | 0.7556 | 0.2002 |  |  |
| Devign | word2vec | 0.2529 | 0.2247 | 0.7265 | 0.4882 |
|  | code2vec | 0.3855 | 0.3576 |  |  |
|  | GGNN | 0.4942 | 0.4683 |  |  |

on slice-level data of the FUNDED (*Wang et al., 2020*) dataset was 6.52% higher than that of vulnerability detectors trained on functional-level data. The proportion of vulnerability-related information contained in fine-grained samples was larger than that of function-level samples, which was conducive to the more effective learning of vulnerability-related features by the DL model.

### Insight

For DL-based vulnerability detectors, a training set with fine-grained code samples is conducive to detecting vulnerabilities. The fine-grained samples make it easier for the DL-based vulnerability detector to learn the characteristics of vulnerabilities.

### Impact of sample similarity (RQ2)

This subsection was to study the impact of sample similarity on DL-based vulnerability detectors. We used word2vec (*Mikolov et al., 2013*), code2vec (*Alon et al., 2019*), and GGNN (*Zhou et al., 2019*) to represent the samples of the three vulnerability datasets (*i.e.*, SySeVR (*Li et al., 2021c*), FUNDED (*Wang et al., 2020*), and Devign (*Zhou et al., 2019*)) as vectors. To better retain important information, we then used PCA (*Wold, Esbensen & Geladi, 1987*) to reduce the dimensionality of the vector to 15 dimensions. Finally, we used T-SNE (*Laurens & Hinton, 2008*) to reduce the dimensionality of the vector to two dimensions. We calculated the inter-class and the intra-class distances for these two-dimensional vectors and represented their inter-class and intra-class similarities. The results are shown in Table 4. The SySeVR (*Li et al., 2021c*) dataset had the highest inter-class similarity and the lowest intra-class similarity. In contrast, the FUNDED (*Wang et al., 2020*) dataset had the lowest inter-class similarity and the highest intra-class similarity across the three representations. The average inter-class distance of SySeVR (*Li et al., 2021c*) was 1.45 times higher than Devign (*Zhou et al., 2019*) and 2.08 times higher than FUNDED (*Wang et al., 2020*). The average intra-class distance of FUNDED (*Wang et al., 2020*) was 1.37 times higher than Devign (*Zhou et al., 2019*) and 2.45 times higher than SySeVR (*Li et al., 2021c*).

**Table 5 The results of the vulnerability detectors trained by the three training sets on the original test set and REVEAL.**

| Test set | Detector | Training set | Accuracy (%) | Precision (%) | Recall (%) | F1-score (%) |
|---|---|---|---|---|---|---|
| Original | VulDeePecker | FUNDED | 63.75 | 53.45 | 51.78 | 52.65 |
| | | SySeVR | 89.52 | 75.62 | 72.37 | 73.59 |
| | | Devign | 58.57 | 68.43 | 60.36 | 64.17 |
| | C2V-BGRU | FUNDED | 52.52 | 54.13 | 42.73 | 47.79 |
| | | SySeVR | 84.22 | 56.35 | 49.52 | 52.68 |
| | | Devign | 53.58 | 52.53 | 46.74 | 49.43 |
| | REVEAL | FUNDED | 48.84 | 49.44 | 48.94 | 49.23 |
| | | SySeVR | 79.05 | 56.82 | 74.60 | 64.42 |
| | | Devign | 66.24 | 47.24 | 65.87 | 55.31 |
| REVEAL | VulDeePecker | FUNDED | 78.74 | 23.78 | 28.93 | 26.41 |
| | | SySeVR | 80.56 | 9.54 | 15.59 | 11.83 |
| | | Devign | 70.08 | 10.56 | 17.58 | 13.19 |
| | C2V-BGRU | FUNDED | 89.05 | 21.56 | 19.35 | 20.39 |
| | | SySeVR | 88.41 | 8.33 | 14.78 | 10.65 |
| | | Devign | 84.23 | 19.72 | 14.88 | 16.96 |
| | REVEAL | FUNDED | 66.26 | 35.89 | 20.36 | 25.98 |
| | | SySeVR | 72.38 | 8.76 | 17.33 | 11.63 |
| | | Devign | 64.05 | 22.35 | 17.45 | 19.59 |

We generated function-level code samples from the three vulnerability datasets (*i.e.*, SySeVR (*Li et al., 2021c*), FUNDED (*Wang et al., 2020*), and Devign (*Zhou et al., 2019*)) according to the input format required by the three function-level vulnerability detectors (*i.e.*, VulDeePecker (*Zhen et al., 2018*), REVEAL (*Chakraborty et al., 2020*), and C2V-BGRU). Since both VulDeePecker and REVEAL used the ratio of 80%: 20% in original papers, we followed this ratio to ensure maximum restoration of vulnerability detectors. For every vulnerability dataset, 80% of the samples were training set and 20% of the samples were original test set. Training sets were used to train DL-based vulnerability detectors, test sets, and the REVEAL (*Chakraborty et al., 2020*) dataset were used to test the effect of the vulnerability detector. The results are shown in Table 5 and Figs. 3 and 4.

From Table 5, we observed that for the vulnerability detection of the original test set, the F1-score of VulDeePecker trained by SySeVR (*Li et al., 2021c*) was 9.42% higher than the F1-score trained by Devign (*Zhou et al., 2019*) and 20.94% higher than the F1-score trained by FUNDED (*Wang et al., 2020*). The F1-score of C2V-BGRU trained by SySeVR (*Li et al., 2021c*) was 3.25% higher than the one trained by Devign (*Zhou et al., 2019*) and 4.89% higher than the one trained by FUNDED (*Wang et al., 2020*). The F1-score of REVEAL (*Chakraborty et al., 2020*) trained by SySeVR (*Li et al., 2021c*) was 9.11% higher than the F1-score trained by Devign (*Zhou et al., 2019*) and 15.19% higher than the F1-score trained by FUNDED (*Wang et al., 2020*). The average F1-score of SySeVR (*Li et al., 2021c*) was 7.26% higher than the F1-score trained by Devign (*Zhou et al., 2019*) and 13.67% higher than the F1-score trained by FUNDED (*Wang et al., 2020*).

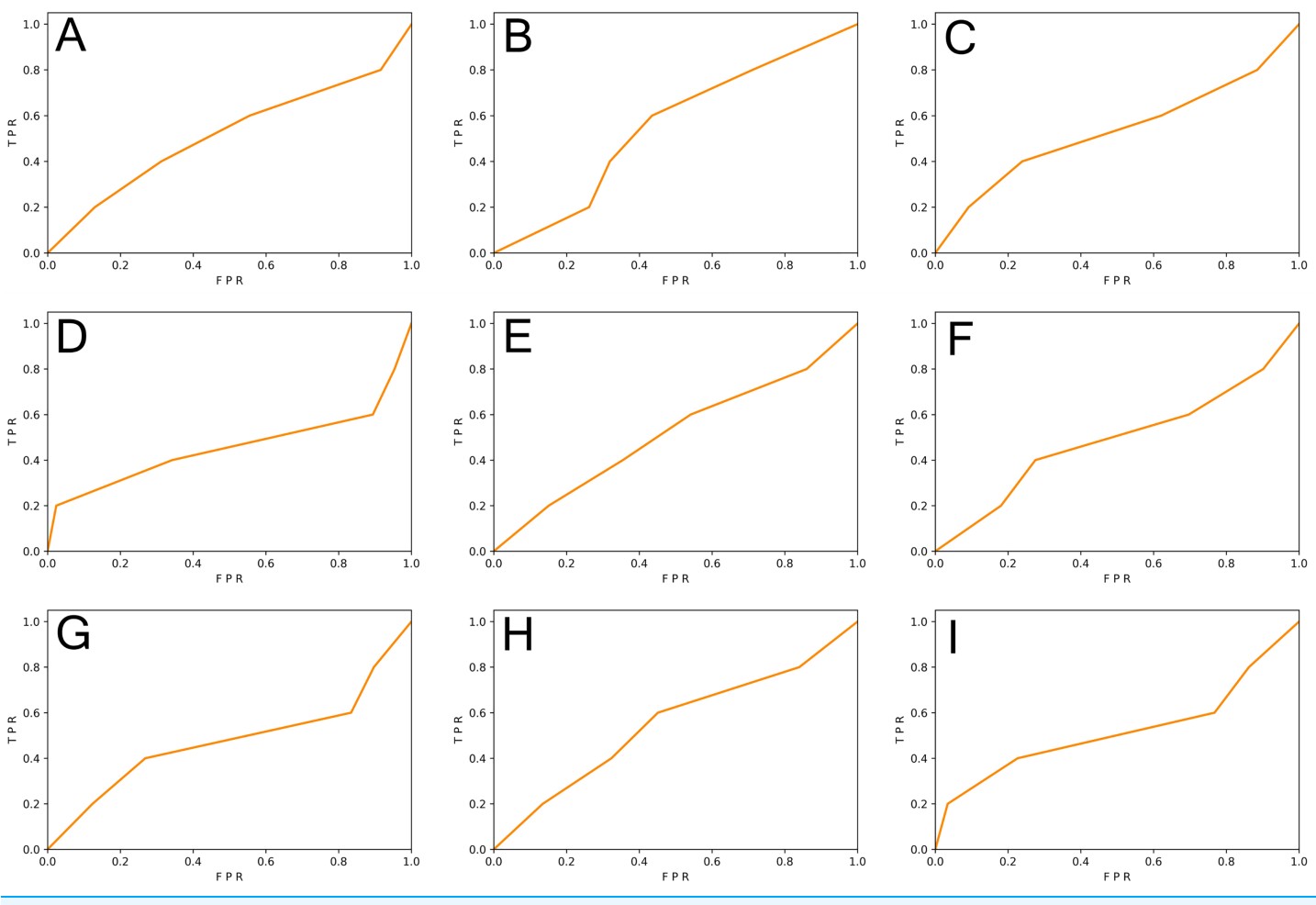

**Figure 3** (A–I) The ROC curve of three vulnerability detectors tested on the original test set.

For the vulnerability detection of the REVEAL (*Chakraborty et al., 2020*) datasets, the F1-score of VulDeePecker trained by FUNDED (*Wang et al., 2020*) was 13.22% higher than the F1-score trained by Devign (*Zhou et al., 2019*) and 14.58% higher than the F1-score trained by SySeVR (*Li et al., 2021c*). The F1-score of C2V-BGRU trained by FUNDED (*Wang et al., 2020*) was 3.43% higher than the one trained by Devign (*Zhou et al., 2019*) and 9.74% higher than the one trained by SySeVR (*Li et al., 2021c*). The F1-score of REVEAL (*Chakraborty et al., 2020*) trained by FUNDED (*Wang et al., 2020*) was 6.39% higher than the F1-score trained by Devign (*Zhou et al., 2019*) and 14.35% higher than the F1-score trained by SySeVR (*Li et al., 2021c*). The average F1-score of FUNDED (*Wang et al., 2020*) was 7.68% higher than the F1-score trained by Devign (*Zhou et al., 2019*) and 12.89% higher than the F1-score trained by SySeVR (*Li et al., 2021c*).

By analyzing the above results, we observed that for the vulnerability detection of the original test set, the dataset with lower inter-class similarity and higher intra-class similarity was better. For the vulnerability detection of the REVEAL (*Chakraborty et al.,*

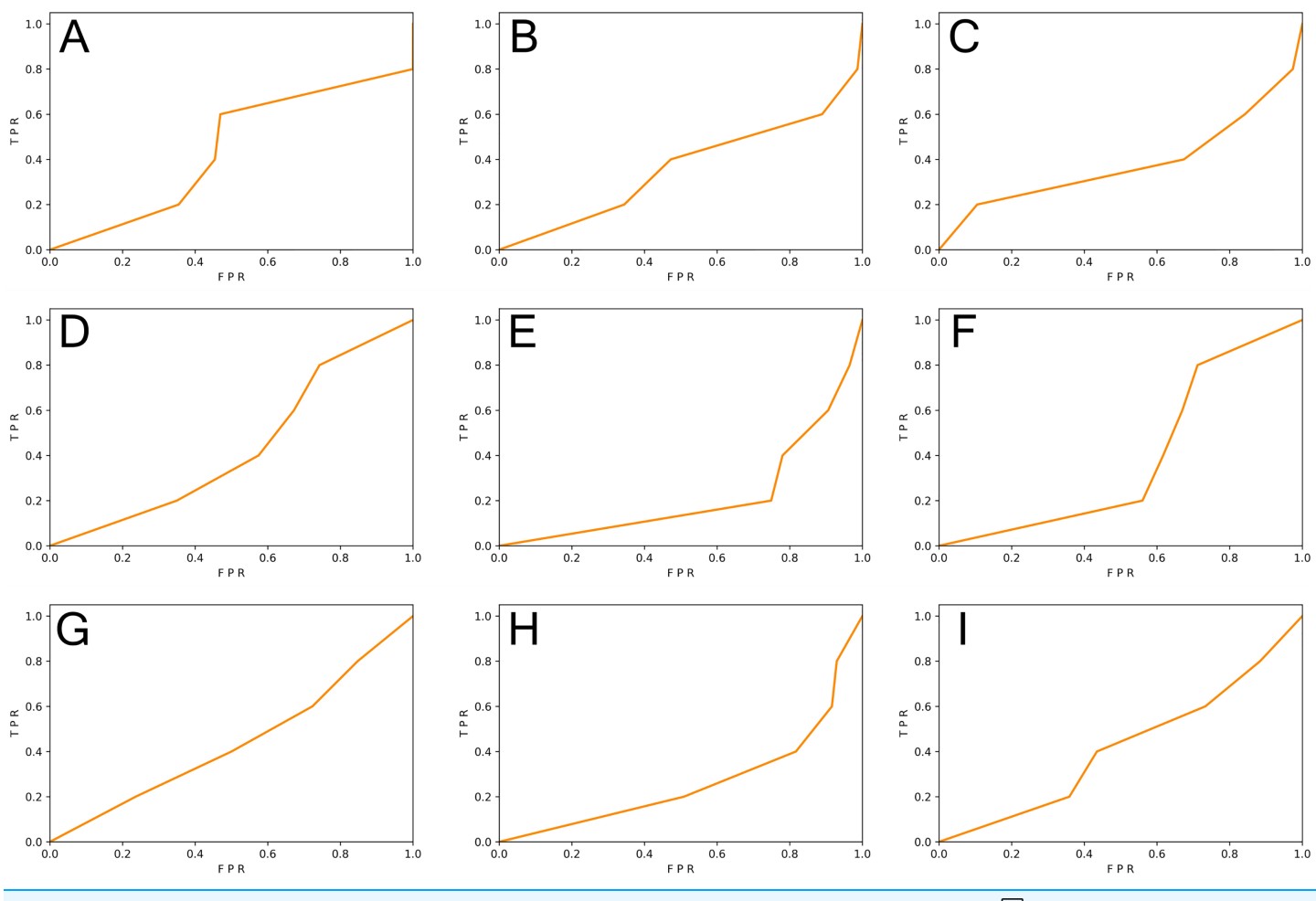

**Figure 4 (A–I) The ROC curve of three vulnerability detectors tested on the REVEAL dataset.**

2020) dataset, the dataset with higher inter-class similarity and lower intra-class similarity was better.

The training set with lower inter-class similarity made it easier for the DL model to learn the difference between the two data classes. For the vulnerability detection of the original test set, the similarity between the test set was also lower. The difference between the vulnerable and non-vulnerable classes was significant and was more conducive to detecting vulnerable samples. However, since the DL model learns different features that are not related to vulnerabilities, the training set with low inter-class similarity will affect the performance of the vulnerability detector when detecting other vulnerability datasets.

The training set with higher intra-class similarity helped the DL model learn the characteristics of the two classes of data. For the vulnerability detection of the original test set, the intra-class similarity of the test set was also higher. The same class of data was slightly different, making it more conducive to detecting vulnerable samples. However, since the higher intra-class similarity meant that the similar data in the dataset had a single feature, the DL model learned features unrelated to the vulnerability. Therefore, for the

**Table 6 Code features values of three vulnerability datasets.**

| Dataset | AvgCyclomalic | AvgEssential | AvgLine (num) | AvgCountInput (num) | AvgCountOutput (num) |
|---------|---------------|--------------|---------------|---------------------|----------------------|
| FUNDED | 15.37 | 8.09 | 100.05 | 8.97 | 17.17 |
| SySeVR | 8.38 | 4.84 | 51.87 | 5.96 | 7.08 |
| Devign | 9.23 | 4.98 | 74.77 | 7.70 | 11.13 |

detection of other vulnerability datasets, the high intra-class similarity of the training set affects the performance of the vulnerability detector.

### *Insight*

For DL-based vulnerability detectors, vulnerability datasets with higher intra-class similarity and lower inter-class similarity are conducive to detecting vulnerabilities in the original test set. Vulnerability datasets with lower intra-class similarity and higher inter-class similarity are conducive to detecting vulnerabilities in other vulnerability datasets. This is because higher intra-class similarity and lower inter-class similarity cause DL-based vulnerability detectors to learn a single feature and features that are unrelated to vulnerabilities.

## Impact of code features (RQ3)

This subsection studied the impact of code features on the DL-based vulnerability detector. We analyzed the samples of vulnerability datasets, extracted and characterized the code features of the vulnerability datasets, and the results are shown in Table 6. We found that the SySeVR (*Li et al., 2021c*) dataset had the lowest complexity, smallest sample size, and minor subroutine calls. The FUNDED (*Wang et al., 2020*) dataset had the highest complexity, largest sample size, and most subroutine calls.

From Tables 5 and 6, we observed that for the vulnerability detection of the original test set, a training set with lower complexity, smaller average sample size, and fewer subroutine calls was better. For vulnerability detection of the REVEAL (*Chakraborty et al., 2020*) dataset, a training set with higher complexity, larger average sample sizes, and more subroutine calls generated better results.

The dataset with low complexity, small sample size, and fewer subroutine calls had a simple structure, and it was easier for the DL models to learn simple inputs. For the vulnerability detection of the original test set, the structure of the test set was also simple, so it was more conducive to detecting vulnerable samples. However, the simple structure meant that the dataset had a single feature, which made it difficult for the DL model to learn complex vulnerability features from the training set. Therefore, for the vulnerability detection of other vulnerability datasets, a training set with low complexity, small sample size, and few subroutine calls will affect the performance of the vulnerability detector.

### *Insight*

For DL-based vulnerability detectors, vulnerability datasets with a simple structure are conducive to detecting vulnerabilities in the original test set, and vulnerability datasets

with a complex structure are more conducive to detecting vulnerabilities in other vulnerability datasets. This is because it is not easy to detect vulnerabilities in complex data, and their complex features can better train the detection ability of DL-based vulnerability detectors.

# DISCUSSION

## Suggestions

Based on our research results, we have the following suggestions for creating and selecting vulnerability datasets. (1) Vulnerability datasets should be collected from the real-world environment. We need to de-duplicate the dataset and remove irrelevant vulnerability information, such as header files and comments. (2) When verifying the feasibility of the DL-based vulnerability detector, the chosen dataset should be relatively simple with less similarity between classes and more significant intra-class similarity under the representation of the DL-based vulnerability detector. (3) When optimizing the DL-based vulnerability detector, the chosen dataset should be more complex with more significant similarity between classes and less similarity within classes under the representation of the DL-based vulnerability detector.

## Limitations

This study has several limitations. First, we used three vulnerability datasets and four DL-based vulnerability detectors for research. Due to the limited number of public vulnerability datasets currently available and the inherent limitations of DL-based vulnerability detectors, more DL-based vulnerability detectors and vulnerability datasets should be used to verify the results in the future. Second, we studied the impact of C/C++ vulnerability datasets on DL-based vulnerability detectors. Future research direction should explore Python/Java/PHP vulnerability datasets. Third, our work was devoted to the existing vulnerability datasets, but we did not conduct in-depth research on how to improve vulnerability datasets. In the future, we will provide a complete improvement plan.

# CONCLUSION

This article focuses on using sample granularity, sample similarity, and code features to study the impact of vulnerability datasets on DL-based vulnerability detectors. Our research found: (1) Fine-grained samples were conducive to detecting vulnerabilities; (2) vulnerability datasets with lower inter-class similarity, higher intra-class similarity, and simple structure helped detect vulnerabilities in the original test set; and (3) vulnerability datasets with higher inter-class similarity, lower intra-class similarity, and complex structure could better detect vulnerabilities in other datasets. We also have given suggestions for creating and selecting vulnerability datasets. During the research process, we found that the quality of vulnerability datasets was essential to DL-based vulnerability detectors. It affected the DL-based vulnerability detectors and played a significant role in guiding the optimization of DL-based vulnerability detectors. The lack of vulnerability datasets restricts the development of DL-based vulnerability detectors. We hope to collect

better vulnerability datasets to study their relationship with vulnerability detectors and lay a foundation for developing DL-based vulnerability detectors.

### Funding

This work was supported by the Natural Science Foundation of Hebei Province under Grant No. F2020201016. The funders had no role in study design, data collection and analysis, decision to publish, or preparation of the manuscript.

### Grant Disclosures

The following grant information was disclosed by the authors:
Natural Science Foundation of Hebei Province: F2020201016.

### Competing Interests

The authors declare that they have no competing interests.

### Author Contributions

- Lili Liu conceived and designed the experiments, performed the experiments, analyzed the data, performed the computation work, prepared figures and/or tables, authored or reviewed drafts of the paper, and approved the final draft.
- Zhen Li conceived and designed the experiments, performed the computation work, authored or reviewed drafts of the paper, and approved the final draft.
- Yu Wen analyzed the data, prepared figures and/or tables, and approved the final draft.
- Penglong Chen performed the experiments, analyzed the data, performed the computation work, prepared figures and/or tables, authored or reviewed drafts of the paper, and approved the final draft.

### Data Availability

The data is available at figshare: Liu, Lili (2021): PCS.zip. figshare. Dataset. https://doi.org/10.6084/m9.figshare.17059688.v1

The code is available at GitHub: https://github.com/liulili0925/peerj-cs.git.

The third-party data and code are available at GitHub:

- https://github.com/HuantWang/FUNDED_NISL
- https://git.io/Jf6IA
- https://github.com/SySeVR/SySeVR/.

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
