# Peer review of "Investigating the impact of vulnerability datasets on deep learning-based vulnerability detectors"

_PeerJ Computer Science, doi:10.7717/peerj-cs.975_

## Round 0.1 · original submission · Major Revisions

Please revise the paper accordingly.

Reviewer 1 ·

Basic reporting

Paper summary:
This paper studies the impact of datasets' distribution on the performance of deep learning-based vulnerability detectors, Three different aspects of vulnerability datasets were investigated, including the dataset granularity (function level or slice level), similarity (inter-class similarity and intra-class similarity), and the code features, such as AvgCyclmotic, AveEssential. In the evaluation part, several datasets and deep learning models are evaluated according to the three aspects, and associated insights were proposed.

Advantages:
+ I think this paper has very good research insights on the deep learning model datasets, which does not draw much attention in previous literature. Meanwhile, the distribution of datasets is in fact very important to deep learning models' performance.

+ The overall experiment design is reasonable and clear.

+ The datasets used in this paper as well as the deep learning models are very new.

Weakness:
I have several concerns about this paper, as summarized below:

+ Overall, the writing of this paper requires improvement. The high-level structures of this paper are relatively clear, but there are flaws:
1) The abstract needs to be more succinct, especially on the method description. Such as "The training set is used to train the DL-based vulnerability detector". Such statements do not need to be in the abstract.
2) In the Experimental Result Section, I think it will be better to switch the order of Table 4 and Table 5, and their introduction. It is more intuitive to present the representation similarity of the datasets first before giving the corresponding evaluation results.
3) In Table 2, I think it will be better to add a column to show the category of the dataset, such as synthesized dataset, manually modified, and open-source software. I understand such information can be found in RELATED WORK, doing so will make the table clear and release the unnecessary burden of readers.
4) Typos such as "Dulnerability Datasets" IN Section EXPERIMENTAL SETUP Line 186.
5) In the INTRODUCTION section, redundant information on "what's deep learning" is given. I think this part could also be more succinct since the knowledge should be already known to the journal readers.

+ Another concern is the motivation to select the three aspects (granularity, similarity, code features) is not clearly presented in the paper. More evidence and motivation should be added to show why these three factors are important for evaluations. In addition, the similarity and code features are in fact characteristics of the raw code datasets, while granularity can be considered as a kind of preprocessing. I think it will be better to discuss them separately.

+ The details of experimental implementation are not given. For example, I was expecting the introduction to experiment platforms, the libraries used, experiment data such as running time.

+ Still in the evaluation part, the sizes of the datasets are not given. Also, the parameters (m, n) mentioned in STEP I (Line 146) are not given and not evaluated.

Experimental design

As shown above.

Validity of the findings

As shown above.

Additional comments

no comment

Reviewer 2 ·

Basic reporting

This paper analyzes the inner connection between DL-based vulnerability detectors and datasets. The paper mainly focuses on the following aspects: fine-grained samples, datasets with lower inner-class similarity, and datasets with higher inter-class similarity and lower intra-class similarity. Although many other points can be explored in this kind of research, this paper grants a good view of the relationship between dataset and detector schemes. Also, this is interesting research, giving guidelines to the related research.

Experimental design

This is the most important part of the paper since the author claims their contributions focus on the evaluation. More effort should be put into this part.

Validity of the findings

The author needs to put more effort into the evaluation parts, as mentioned above. Some suggestions:
1. using recall rate and an F score is good, but ROC curve, EER, and accuracy are also necessary to evaluate various schemes. Since this part is the paper's focus, authors are suggested to provide detailed evaluation results in the paper.

2. How did you divide the dataset? What is the training-testing ratio? This will also impact the detectors' performance.

3. What was information loss when you applied PCA to the dataset? and what is the energy and time consumption when you remove the PCA?

4. Instead of using intra- or extra-class similarity, authors are suggested to evaluate the entropy difference between data samples.

Reviewer 3 ·

Basic reporting

1- This paper has not reached to the acceptable level for publication because of lacks originality and novelty.
2- The spell-checks, grammatical and writing style errors of the paper must be improved.

Experimental design

No comment

Validity of the findings

No comment

Additional comments

1- For readers to quickly catch the contribution in this work, it would be better to highlight major difficulties and challenges, and your original achievements to overcome them, in a clearer way in abstract and introduction.
2- Some references are too old and please add at least five references within the past one year for related work section.
3-There are unsatisfactory organization and writing in the "Conclusion" section. The authors must rewrite and reorganize this section contents.
4- The highlights are not stressed in the paper, and the innovation of the paper weakness.
5- There are some mistakes in the style of English writing in the text which are to be revised/corrected carefully.

---

## Round 0.2 · accepted · Accept

From the comments of the reviewer, I recommend to accept this paper.

# Reviewer 1 ·

Basic reporting

The authors solved the issues mentioned in the review comments well. This version looks good to me.

Experimental design

See above.

Validity of the findings

See above.